# Fatigue in Kidney Transplantation: A Systematic Review and Meta-Analysis

**DOI:** 10.3390/diagnostics11050833

**Published:** 2021-05-05

**Authors:** Maurizio Bossola, Maria Arena, Federica Urciuolo, Manuela Antocicco, Gilda Pepe, Giovanna Elisa Calabrò, Claudia Cianfrocca, Enrico Di Stasio

**Affiliations:** 1Dipartimento di Scienze Mediche e Chirurgiche, Università Cattolica del Sacro Cuore, 00168 Roma, Italy; maurizio.bossola@gmail.com (M.B.); maria.arena90@virgilio.it (M.A.); federicaurciuolo@gmail.com (F.U.); gilda.pepe@policlinicogemelli.it (G.P.); 2Fondazione Policlinico Universitario A. Gemelli IRCCS, 00168 Roma, Italy; manuela.antocicco@policlinicogemelli.it; 3Divisione di Nefrologia, Università Cattolica del Sacro Cuore, 00168 Roma, Italy; 4Dipartimento di Geriatria, Università Cattolica del Sacro Cuore, 00168 Roma, Italy; 5Dipartimento di Scienze della Vita e Sanità Pubblica, Sezione di Igiene, Università Cattolica del Sacro Cuore, 00168 Roma, Italy; giovannaelisa.calabro@unicatt.it; 6Dipartimento di Biomedicina e Prevenzione, Università degli Studi di Roma “Tor Vergata”, 00133 Roma, Italy; c2211@inwind.it; 7Dipartimento di Scienze Biotecnologiche di Base, Cliniche Intensivologiche e Perioperatorie, Università Cattolica del Sacro Cuore, 00168 Roma, Italy

**Keywords:** kidney transplant recipients, patients on hemodialysis, fatigue, assessment

## Abstract

Fatigue is still present in up to 40–50% of kidney transplant recipients (KTR), the results of studies comparing the prevalence among patients on hemodialysis (HD) and KTR led to conflicting results. Fatigue correlates include inflammation, symptoms of depression, sleep disorders and obesity. Fatigue in KTR leads to significant functional impairment, it is common among KTR poorly adherent to immunosuppressive therapy and is associated with a serious deterioration of quality of life. The following databases were searched for relevant studies up to November 2020: Medline, PubMed, Web of Science and the Cochrane Library. Several studies have compared the prevalence and severity of fatigue between KTR and hemodialysis or healthy patients. They have shown that fatigue determines a significant functional deterioration with less chance of having a paid job and a significant change in quality of life. The aim of the review is to report methods to assess fatigue and its prevalence in KTR patients, compared to HD subjects and define the effects of fatigue on health status and daily life. There is no evidence of studies on the treatment of this symptom in KTR. Efforts to identify and treat fatigue should be a priority to improve the quality of life of KTR.

## 1. Introduction

Fatigue is a complex phenomenon that involves many aspects of existence and is determined by physical, psychological and emotional components. Fatigue can be also described as a condition that causes distress and decreases ability to function due to a lack of energy. Fatigue is expressed by humans in different ways, such as saying they feel tired, weak, exhausted, weary, worn-out, heavy or slow. Health professionals may use terms such as asthenia, fatigue, lassitude, prostration, exercise intolerance, lack of energy and weakness to describe fatigue [1,2].

It is generally believed that renal transplantation improves the quality of life of kidney transplant recipients (KTR) compared to dialysis treatments [3,4,5,6]. However, a recent meta-analysis has shown that the difference in health-related quality of life measured by the SF-36 instrument between KTR and hemodialysis patients was significantly reduced after controlling for age and diabetes [7]. Accordingly, others have demonstrated that the differences in perceived health status between KTR and dialyzed patients were based mainly on the selection progress [8]. In addition, after kidney transplantation, many patients still face the debilitating and prostrating symptom of fatigue [9,10]. Unfortunately, this symptom is often underestimated as demonstrated by the observation that among the 63 KTR patients (out of 106) experiencing fatigue, only 8 (13%) had complaints of fatigue documented in medical records [9].

Little is known about fatigue in KTR. The aim of the present review is to report the methods to assess fatigue in KTR, the prevalence of fatigue in KTR in comparison with healthy individuals and hemodialysis patients, to define the effects of fatigue on the health status and on the daily living, in order to summarize the demographic, clinical and laboratory variables associated with its presence, thus indicating possible therapeutic strategies.

## 2. Materials and Methods

The following databases were searched for relevant studies up to November 2020: Medline, PubMed, Web of Science and the Cochrane Library. The search terms and mesh headings included “fatigue” AND “kidney” AND “transplantation” OR “transplant” AND “end-stage renal disease” AND “hemodialysis”. Reference lists of relevant studies and previous systematic reviews were manually searched for additional articles. Abstracts presented at international congresses and personal communication were also considered. Studies were eligible for inclusion if they were English language papers published in a peer-reviewed journal and met the following inclusion criteria: (1) primary research studies in adult patients (over 18 years of age), (2) included patients with end-stage renal disease on chronic hemodialysis or peritoneal dialysis who underwent renal transplantation and (3) investigated fatigue. 

We followed PRISMA guidelines in the redaction of the present review. PRISMA flow diagram is reported in Figure 1.

According to PICOS criteria, we analyzed: Population = Nephrological patients; Intervention = Kidney Transplantation; Comparison = Hemodialysis; Outcome = Fatigue.

The primary outcome of the review is to determine fatigue prevalence (odds ratio) in hemodialysis (HD) compared to KTR patients in different studies using various assessment scales. The heterogeneity of fatigue measurements scales, applied in each study, is the major bias in order to compare results from different protocols; moreover, some studies have not reported, explicitly, the fatigue prevalence. Therefore, on the basis of the Gaussian or percentile data distribution declared in each manuscript, we extrapolated the number of subjects suffering fatigue for HD or KTR subgroups using the fatigue definition cut-off point for each applied scale (for debating values the scale midpoint was used as cutoff). Finally, a meta-analysis on the prevalence of fatigue (number of subjects suffering fatigue in respect to total enrolled patients) in HD or KTR patients was carried out and the OR (M-H random effect), weight of the single study and heterogeneity parameters (Tau, Chi^2^, p, I^2^ and Z and p for overall effect) were reported.

## 3. Results

### 3.1. Measurement of Fatigue in KTR and HD Patients

Fatigue in kidney recipients has been evaluated with various instruments. Noteworthy, most of these were not specifically designed, manufactured and validated for kidney recipients. The Kidney Transplant Questionnaire is the only specific fatigue scale dedicated to kidney transplant patients. 

*KTQ-fatigue subscale.* The Kidney Transplant Questionnaire is a 25-item questionnaire that includes five domains or subscales, i.e., physical symptoms (based on six items), fatigue (based on five items), uncertainty/fear (based on four items), appearance (based on four items) and emotional (based on six items). A mean score ranging from one to seven is reported for each of the five subscales, with higher scores representing better functioning, well-being, or fewer problems [11,12,13].

*Multidimensional Fatigue Inventory (MFI-20).* The MFI-20 was developed by a Dutch group in 1995 to measure fatigue severity [14]. The MFI contains 20 statements organized into five dimensions of fatigue with four statements each (general fatigue, physical fatigue, reduced activity, reduced motivation and mental fatigue). The response-scale has five choices from agreement “yes, that is true” to disagreement “no, that is not true”. A global fatigue score combining the five results ranges from 20 to 100, with higher scores indicating higher levels of fatigue. 

*Checklist Individual Strength-20 (CIS-20).* CIS-20 consists of 20 statements and provides a total fatigue score and scores for four components of fatigue: subjective experience of fatigue (SEF; eight items), reduced concentration (CON; five items), reduced motivation (MOT; four items) and reduced physical activity level (PA; three items) [15]. Respondents use a seven point rating scale (1, yes, that is true, to 7, no, that is not true). A total score above 76 is considered high [16].

*SF-36 Vitality subscale.* The four-item SF-36 Vitality Scale has been used as a proxy measure for fatigue. Using a Likert-type scale, patients report their agreement with two positively scored items (“Did you feel full of life?” and “Did you have a lot of energy?”) and two negatively scored items (“Did you feel worn out?” and “Did you feel tired?”). Like the other SF-36 scales, the vitality scores are standardized according to U.S. population norms, and scores range from 0 to 100, with higher scores reflecting greater vitality in the past month. The SF-36 Vitality Scale explores both fatigue and a related concept, energy level [17].

*Fatigue Symptom Inventory (FSI)*. The FSI is a 14-item measure that assesses the severity, frequency, and diurnal variation of fatigue, as well as its perceived interference with quality of life. Severity is measured using four separate items that assess most, least and average fatigue in the past week as well as current fatigue. Frequency is measured using two separate items that assess the number of days in the past week that respondents felt fatigued as well as the portion of each day on average they felt fatigued. Diurnal variation is measured using a single item that provides descriptive information about daily patterns of fatigue. Perceived interference is measured using seven separate items that assess the degree to which fatigue in the past week was judged to interfere with general level of activity, ability to bathe and dress, normal work activity, ability to concentrate, relations with others, enjoyment of life and mood [18].

*Kidney Disease Quality of Life-Short Form (KDQOL-SF-tm).* Hays et al., in 1994, developed a 43 disease-specific item tool for individuals with end-stage renal disease undergoing dialysis. Together with the generic SF-36 these items constitute the KDQOL-SF-tm questionnaire.

*Functional Assessment of Chronic Illness Therapy-Fatigue questionnaire (FACIT-F).* The scale encompasses physical, functional, emotional and social aspects of fatigue. It uses 13-item Functional Assessment of Chronic Illness Therapy-Fatigue questionnaire, which was scored from 0 to 52, with a higher score indicating lower levels of fatigue with an assessment of fatigue over 7 days using a five-point Likert scale ranging from “Not at all” to “Very much”. This scale has excellent internal consistency, test–retest reliability and has been validated in many populations including the general US population and many chronic disease including kidney disease patients [19]. 

### 3.2. Prevalence of Fatigue in KTR and HD Patients

The prevalence of fatigue in KTR has been reported in few studies and varies between 22% and 63%. Usually, kidney transplantation is thought to improve the burden symptom of patients with end-stage renal disease in chronic hemodialysis. If this also occurs for the fatigue symptom remains unclear. In fact, the results of studies comparing the prevalence and severity of fatigue among patients on hemodialysis and KTR led to conflicting results and are shown in Table 1.

Tomasz et al., comparing 61 patients in chronic hemodialysis and 83 KTR through the World Health Organization Quality of Life Questionnaire (WHOQOL-100) questionnaire, found that the domain of energy and fatigue was significantly better in KTR (14.5 ± 2.8) than in patients in hemodialysis (12.4 ± 2.8; *p* < 0.0001) [20]. In the larger study by Kovacs et al. which included 888 KTR and 187 hemodialysis patients, the score of the SF-36 vitality subscale was significantly higher in KTR (median (25th–75th pc): 70 (53–88)) than in hemodialysis patients (median (25th–75th pc): 60 (42–77); *p* < 0.001). However, in the multivariate regression analysis, the dominance of energy fatigue was not independently associated with any of the renal replacement therapies [21]. Finally, Rodrigue et al. showed that hemodialysis patients (*n* = 100), compared to KTR (*n* = 100), had, with Fatigue Symptom Inventory, a higher frequency of fatigue (4.3 ± 2.5 vs. 3.8 ± 2.6 days in past week felt fatigued; *p* = 0.16), a higher severity (3.6 ± 2.8 vs. 2.7 ± 2.6; *p* = 0.02) and an alteration in fatigue. In addition, the scores of general, physical, emotional and mental fatigue at the Multidimensional Fatigue Symptom Inventory were significantly lower in KTR than in hemodialysis patients [22]. It is interesting to note that in the longitudinal study of Kostro et al., which included 44 patients who were evaluated for quality of life through the KDQOL-SF questionnaire during dialysis and after 12 months after kidney transplantation, the energy/fatigue domain improved significantly [23].

Van Sandwijk et al. conducted a study in HD patients, kidney transplant recipients, patients with a hematological malignancy either receiving chemotherapy or in remission and healthy controls. They demonstrated that HD patients and hematological patients undergoing chemotherapy were more frequently severely fatigued compared with KTR, hematological patients in remission and healthy controls, but still had a lower overall QoL than healthy controls, comparable to hematological patients in remission [16]. In the study of Iqbal et al., the energy/fatigue score on the KDQOL-SF-36 scale were significantly better in KTR (66 ± 11) than in patients on chronic hemodialysis (40 ± 7) [24].

Another longitudinal study showed that patients with kidney transplant, unlike patients who did not receive kidney transplantation, reported a significant improvement of fatigue (FACIT-F score from 38.1 in pre-transplant period to 42.7 after transplant; *p* = 0.02), that remained significant adjusting for age, sex and BMI [19].

In the large multicenter study of Mouheli et al., which included 1424 French kidney transplant patients, fatigue was evaluated on the SF-36 viability scale and was found to be associated with comorbidities evaluated by Charlson’s comorbidity index significantly [25].

Studies reporting comparison between fatigue in HD and KTR subjects are shown in the Forrest plot of Figure 2. Fatigue prevalence in each subgroup was obtained from original data or frequency extrapolation using fatigue score and data distribution of each specific study. A higher rate (OR from 1.29 to 126.00) of fatigue was observed in HD compared to KTR subjects in each study (total OR 3.47 (1.72–6.97)), heterogeneity I^2^ = 80%. The extreme high OR value observed for Iqbal et al. data are probably due to the small reported sample size.

### 3.3. Variables Associated with Fatigue in KTR

The causes of fatigue in KTR are unknown. This is due to the lack of longitudinal studies. In addition, there are few cross-sectional studies that have assessed the variables associated with fatigue in KTR (Table 2).

Ujszaszi et al., studying the correlation between protein-energy malnutrition and quality of life, found that the Malnutrition-Inflammation Score (MIS) was significantly associated with the energy/fatigue domain and other quality of life domains (KDQOL-SF questionnaire) [27].

The recent study by Chan et al. has shown that physical and mental fatigue in KTR, which are significantly increased compared to the control of healthy individuals, does not seem to be related to muscle and cardiovascular factors, but rather seems to be driven by an increased perception of effort during exercise (defined as how much one feels as if one’s body is working hard) [26].

Various dimensions of fatigue assessed through the MFI-20 (general fatigue, physical fatigue, reduced activity and reduced motivation) in KTR have been shown to be independently associated with chronic inflammation [9]. Chronic inflammation is common in KTR and similar in frequency and severity to inflammation in patients with chronic renal disease [27,28,29,30]. 

Fatigue in KTR is significantly associated with symptoms of depression and/or sleep disorders [10,22]. The study of Goedendorp et al. showed that these symptoms were more strongly related to severe fatigue (odds ratio 9.70 and 1.02 respectively; *p* ≤ 0.001 and = 0.013). However, depressive symptoms could not completely explain the presence of severe fatigue. In fact, in this study, of the 39% severely fatigue recipients more than two thirds did not have clinically relevant symptoms of depression. It is interesting to note that such fatigue-related variables (inflammation, depression and sleep disorders) in KTR are found also and commonly in dialysis patients [10].

Finally, two studies have shown that fatigue in KTR is significantly associated with obesity [10,22].

Noteworthy, living donor recipients transplanted less than or equal to five years old reported significantly less fatigue than deceased donor recipients on all subscales of the MFI-20 except for mental fatigue and reduced motivation at univariate analysis [31]. However, after adjustment for age, sex and educational level, differences in fatigue between the two groups of recipients only showed a trend toward significance for the subscales general and mental fatigue. In addition, recipients who had been living and deceased transplanted for more than five years did not differ in reported fatigue.

A small, randomized trial compared KTR to standard maintenance immunosuppression, which includes prednisone therapy, and rapid steroid replacement KTR with interleukin-2 receptor inhibitor. This trial showed that prednisone withdrawal was associated with a significant improvement of fatigue assessed through the SF-36 Vitality Subscale [32]. The study of Lee et al. has demonstrated that KTR with low adherence to immunosuppression had a significantly higher occurrence and distress of symptoms than patients with high or medium adherence after adjusting for a number of covariates and that the most common symptom both in terms of occurrence (96.4%) and distress (91.1%) among poorly adherent KT recipients was tiredness. This is an important finding considering that poor adherence to immunosuppressive agents increases the risk of rejection [33].

### 3.4. Effects of Fatigue in KTR 

Fatigue has some deleterious effects in KTR, as shown in Table 3:

A study has shown that fatigue leads to severe and significant functional deterioration, both when considered globally (Sickness Impact profile: 950 ± 707 in case of severe fatigue vs. 242 ± 274 in non-severely fatigued KTR) or if analyzed at the level of individual domains such as sleep and rest (109 ± 90 vs. 27 ± 39), home (114 ± 108 vs. 31 ± 46), mobility (63 ± 105 vs. 6 ± 23), social interaction (192 ± 216 vs. 42 ± 90), walking (84 ± 111 vs. 17 ± 41), leisure activities (94 ± 81 vs. 23 ± 40), alert behavior (950 ± 707 vs. 62 ± 125) and work limitations (164 ± 170 vs. 62 ± 125) [10];

It is significantly associated with a severe deterioration of quality of life [9,25,26];

Fatigue can also reduce adherence to immunosuppressive agents. Lee et al. demonstrated that KTR with low adherence to immunosuppression had a significantly higher incidence and distress of symptoms than patients with high or medium adherence after adjustment for a number of covariates and that the most common symptom in terms of both incidence (96.4%) and anxiety (91.1%) among recipients of KTR with poor adherence to therapy was fatigue [33]. Similarly, a large Chinese study showed that hypertension, hair loss and tiredness were the three most painful symptoms presenting in KTR and were negatively associated with poor adherence to immunosuppressive drugs [34]. This is an important finding considering that poor adherence to immunosuppressive agents increases the risk of rejection and mortality [35].

### 3.5. Fatigue and Mortality in KTR 

Unlike patients on hemodialysis [36], it is not known whether fatigue is a risk factor for mortality in KTR. In fact, no one has specifically measured the effect of prevalence and severity of fatigue on survival in KTR. However, it is noteworthy that the deterioration of quality of life has been shown to increase the risk of mortality and graft failure in KTR and that the risk remained significant after adjustment for sociodemographic and clinical risk factors [37].

### 3.6. Therapy

The studies that have investigated therapeutic strategy for fatigue in KTR are very few, as detailed in the Table 4. It is well known that exercise and physical activity can reduce the prevalence and severity of fatigue in patients with end-stage renal disease in chronic hemodialysis [38,39,40,41] and other chronic diseases [42,43,44,45,46]. A small pilot study that included 21 kidney recipients and 5 liver recipients, who performed 3 aerobic and strengthening exercise sessions per week for 1 year, reported a significant improvement in the vitality domain of the SF-36 questionnaire [47]. Similarly, in the prospective randomized trial of Senthil Kumar et al., patients randomized for 12 weeks improved functional capacity, muscle strength and fatigue levels [48].

Recently, a randomized controlled trial has shown that plantar reflexology significantly improved fatigue after kidney transplantation compared to controls [49].

A single arm study showed that vitamin D3 (cholecalciferol) 800 IU/d supplementation significantly improved fatigue in kidney transplant patients [50].

## 4. Conclusions

Fatigue is a frequent and underestimated symptom of kidney transplant recipients. However, a significant lower rate of fatigue is observed in KTR patients compared with HD ones. The mechanisms underlying this symptom in KTR patients are essentially unknown. Fatigue is significantly associated with symptoms of depression and/or sleep disorders, is extremely debilitating and significantly impairs the quality of life while it remains unknown if it increases the risk of mortality. There is some evidence that fatigue may be improved by exercise. However, adequate and urgent studies are needed to define the cause and appropriate treatments of fatigue in KTR.

## Figures and Tables

**Figure 1 diagnostics-11-00833-f001:**
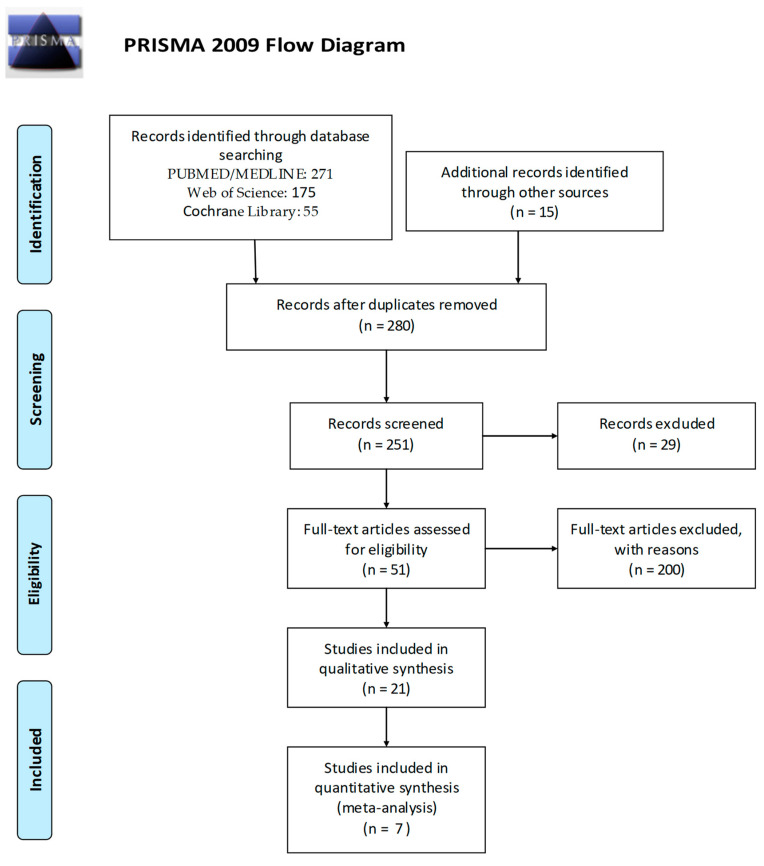
PRISMA flow diagram.

**Figure 2 diagnostics-11-00833-f002:**
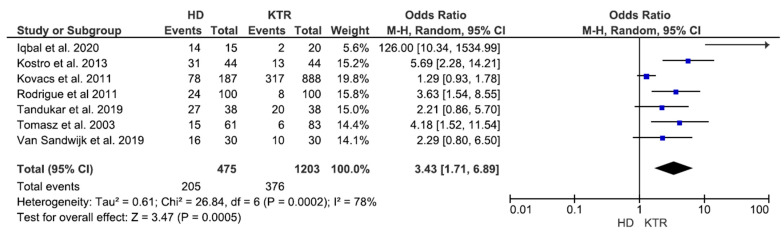
Forrest plot of studies reporting comparison between fatigue in HD and KTR subjects.

**Table 1 diagnostics-11-00833-t001:** Comparison of fatigue between patients on chronic hemodialysis and kidney transplant recipients. CIS, Checklist Individual Strength; FACIT-F, Functional Assessment of Chronic Illness Therapy-Fatigue questionnaire; FSI, Fatigue Symptom Inventory; KDQOL-SFTM, Kidney Disease Quality of Life Short Form; KDQOL-SF-36TM, Kidney Disease Quality of life 36-item short form; MFSI-SF, Multidimensional Fatigue Symptom Inventory; WHOQOL-100, World Health Organization Quality of Life Questionnaire.

Author	Type of Study	HD Patients (*n*)	KTR Patients (*n*)	Measurement	HD Mean ± SD or Median [25th–75th pc]	TX Mean ± SD or Median [25th–75th pc]	*p*	Fatigue Cut-Off	Main Outcome
Tomasz et al. 2003 [20]	Cross-sectional	61	83	WHOQOL-100	12.4 ± 2.8	14.5 ± 2.8	<0.001	<10 Scale Mid-point	Significant differences in energy and fatigue domain
Kovacs et al. 2011 [21]	Cross-sectional	187	888	SF-36 vitality subscale	60 (42–77)	70 (53–88)	<0.001	≤50	Significant differences in SF-36 vitality subscale score
Rodrigue et al. 2011 [22]	Cross-sectional	100	100	FSI, MFSI-SF	21.2 ± 21.5	9.7 ± 19.3	<0.001	>36 Scale Mid-point	Fatigue frequency, severity and disruptiveness higher in pre-transplant patients
Kostro et al. 2013 [23]	Longitudinal	44	44	KDQOL-SFTM	41 ± 18	60± 17	<0.001	≤50	Energy/fatigue domain decrease significantly with kidney transplantation
Van Sandwijk et al. 2019 [16]	Cross-sectional	30	30	CIS	53.3%	33.3%	<0.001	≥40	Prevalence of severe fatigue was higher in HD patients (than in KTR
Tandukar et al. 2019 [19]	Longitudinal	38	39	FACIT	38.1 ± 9.3	42.7 ± 8.8	0.020	≤43	Fatigue improved after kidney transplant
Iqbal et al. 2020 [24]	Cross-sectional	15	20	KDQOL-SF-36TM	40 ± 7	66 ± 11	0.001	≤50	KTR had higher scores of energy/fatigue than hemodialysis patients

**Table 2 diagnostics-11-00833-t002:** Variables associated with fatigue in kidney transplant recipients (KTRs). FSI, Fatigue Symptom Inventory; KDQOL-SFTM, Kidney Disease Quality of Life Short Form; MFI-20, Multidimensional Fatigue Inventory-20; CIS, Checklist Individual strength.

Author	KTR (*n*)	Measurement	Associations
Rodrigue et al., 2011 [9]	100	FSI	Depressive symptoms, sleeping problems, obesity
Ujszaszi et al., 2012 [10]	100	KDQOL-SFtm	MIS significantly associated with the energy/fatigue domain of the KDQOL-SF questionnaire
Chan et al., 2013 [22]	106	MFI-20	Inflammation, decreased estimated glomerular filtration rate and reduced lean tissue index, inferior sleep quality, anxiety and depression
Goedendorp et al., 2013 [25]	151	Subscale fatigue of CIS	Depressive symptoms, sleeping problems, obesity
Chan et al., 2016 [26]	55	MFI-20	Physical fatigue correlated positively with perception of exertion at the Borg RPE Scale
Mouheli et al., 2018 [27]	1247	SF-36 Vitality scale	Charlson comorbidity score, treatment with antidepressant

**Table 3 diagnostics-11-00833-t003:** Effects of fatigue in KTRs. MFI-20, Multidimensional Fatigue Inventory-20; CIS, Checklist Individual strength.

Author	KTR (*n*)	Measurement	Effects
**Quality of life**			
Chan et al., 2013 [9]	106	MFI-20	All fatigue dimensions significantly and inversely correlated with QOL (*p* < 0.001 for all associations).
Chan et al., 2016 [26]	55	MFI-20	Physical fatigue correlated closely with SF-36 total score, SF-36 physical health summary score, and SF-36 mental health summary score.
**Immunosuppression adherence**			
Lee et al., 2015 [33]			Fatigue associated with lower adherence to immunosuppression
**Functional impairment**			
Goedendorp et al., 2013 [10]	151	Subscale fatigue of CIS	Severely fatigued recipients experienced significantly and largely more functional impairments than nonseverely fatigued recipients

**Table 4 diagnostics-11-00833-t004:** Treatment of fatigue in KTR. CIS, Checklist Individual Strength.

Author, Year	Type of Study	Duration of Study	Type of Intervention	Patients (*n*)	Results
Roi et al., 2014 [47]	Prospective, pre/post	12 months	3 sessions per week of aerobic and strengthening exercises	21	Significant improvement of the vitality domain of the SF-36 questionnaire (*p* < 0.05)
Senthil Kumar et al., 2020 [48]	Randomized, controlled	12 weeks	Either routine care vs. exercise training	104	Fatigue score by 0.784 and 1.781 in the control and the study group (SG), respectively, significantly more in the SG
Samarehfekri et al., 2020 [49]	Parallel Randomized Controlled Trial	11 days	Control group: no reflexologyTreatment group: foot reflexology for 30 min once a day for three consecutive days	50	Fatigue improvement was significantly higher (*p* < 0.0001) in the treatment group than in the control group
Han et al., 2017 [50]	Prospective, pre/post	3–9 months	Vitamin D3 supplementation (cholecalciferol) 800 IU/d	60	Subscale fatigue of the CIS significantly improved (*p* = 0.007)

## Data Availability

Data supporting our results can be found in references cited in the study.

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
