# Peer review of "Fatigue in Kidney Transplantation: A Systematic Review and Meta-Analysis"

_diagnostics, 2021, doi:10.3390/diagnostics11050833_

Round 1

Reviewer 1 Report

The authors produced a systematic review and a meta-analysis to evaluate the difference between patients undergoing renal transplantation and patients not transplanted and still undergoing hemodialysis. The topic is quite interesting, but there are some methodological errors.

Major:

1) In Prisma 2009 Flow Diagram, there is no record of how many articles were obtained from the search in each database (Pubmed, Web of Science, Medline and Cochrane Library). This search makes the reproducibility of the work difficult. The authors should write how many results from every database were obtained. Besides, 15 records were obtained by other sources. What are these sources? They should indicate them in the Materials and Methods section.

2) lines 69-71: the PICO criteria are entirely wrong because the authors' hypothesis is to compare kidney transplant patients with non-transplant patients undergoing hemodialysis. The PICO criteria should look like this:
Population = Nephrological patients
Intervention = Kidney Transplantation
Comparison = Hemodialysis
Outcome = Fatigue

3) lines 57-60: following the hypothesis of the authors, the search terms could be: ("Kidney Transplantation" [Mesh]) AND "Fatigue" [Mesh] for Pubmed Database. I recommend reformulating the search using Mesh terms. Suppose the authors want to compare the population of transplanted patients with hemodialysis patients, as stated in line 26. In that case, they do not have to consider patients on peritoneal dialysis (line 60). Please, clarify this point.

4) Table 2: Comparing fatigue between KTR and the healthy subject was not discussed in the introduction methodology section. PICO criteria are different in this case; the Comparison is the healthy population. The authors should give a quantitative analysis of fatigue comparison between KTR and healthy subjects, and if it is not possible, they should remove this table.

5) Authors should write a more precise conclusion. Do transplant patients have significantly lower fatigue rates than hemodialysis patients? Do transplant patients have significantly higher fatigue rates than healthy subjects? What are symptoms significantly associated with the fatigue rate? Does the fatigue rate significantly increase mortality in transplant patients or not? Lines 327 and 328 are extremely vague. Do the therapies and interventions shown in table 5 significantly reduce the fatigue rate in transplant patients? The authors should summarize these points in the conclusions.

Minor:
1) line 23: "healthy patients" should be changed with "healthy subjects."

2) Table 1. and line 165 Kostro et al., in this study, HD/Tx patients are 44, not 69.

3) line 292: the reference is missing

Reviewer 2 Report

The authors focused on one symptom - fatigue  in kidney transplant recipients and conducted systematic review with metaanalysis.

The paper is properly written with adequate  methodology used.

Below are points for clarification/changes:

1. Title should be as one sentence. It is obvious that systematic review is an update for limited time. I suggest :  Fatigue in kidney transplantation a systematic review and metanalysis 

2.Introduction
line 34 and 37 - contains underlined words with hyperlink (cancer.org) - it should be restored to normal style.
2.Results
the interquartile range (IQR) is a measure of statistical dispersion, being equal to the difference between 75th and 25th percentiles. Sometimes IRQ is expressed as Q3-Q1 (70-30) which is more informative than single value. I suggest to unify values as mean+/- SD or IQR.

I am surprised that clinical medicine paper is condidered in more laboratory journal. Editor should reconsider if this suits the journal profile.

Round 2

Reviewer 1 Report

I thank the authors for making the changes. I consider these changes sufficient and comprehensive for publication. I congratulate the authors for their work.
Best wishes